# Dehydration of Fructose to 5-HMF over Acidic TiO_2_ Catalysts

**DOI:** 10.3390/ma13051178

**Published:** 2020-03-06

**Authors:** Maria Luisa Testa, Gianmarco Miroddi, Marco Russo, Valeria La Parola, Giuseppe Marcì

**Affiliations:** 1Istituto per lo Studio dei Materiali Nanostrutturati, ISMN-CNR, Via Ugo La Malfa 153, 90146 Palermo, Italy; gianmarco.miroddi@gmail.com (G.M.); marco.russo@ismn.cnr.it (M.R.); valeria.laparola@cnr.it (V.L.P.); 2Dipartimento di Ingegneria, Viale delle Scienze, edificio n° 6,Università di Palermo, 90128 Palermo, Italy; giuseppe.marci@unipa.it

**Keywords:** HMF, solid acid catalysts, titania, biomass, hydrothermal dehydration

## Abstract

Different solid sulfonic titania-based catalysts were investigated for the hydrothermal dehydration of fructose to 5-hydroxymethylfurfural (5-HMF). The catalytic behavior of the materials was evaluated in terms of fructose conversion and selectivity to 5-HMF. The surface and structural properties of the catalysts were investigated by means of X-ray diffraction (XRD), N_2_ adsorption isotherms, thermo-gravimetric analysis (TGA), X-ray photoelectron spectroscopy (XPS) and acid capacity measurements. Special attention was focused on the reaction conditions, both in terms of 5-HMF selectivity and the sustainability of the process, choosing water as the solvent. Among the various process condition studied, TiO_2_-SO_3_H catalyzed a complete conversion (99%) of 1.1M fructose and 5-HMF selectivity (50%) and yield (50%) at 165 °C. An important improvement of the HMF selectivity (71%) was achieved when the reaction was carried out by using a lower fructose concentration (0.1M) and lower temperature (140 °C). The catalytic activities of the materials were related to their acid capacities as much as their textural properties. In particular, a counterbalance between the acidity and the structure of the pores in which the catalytic sites are located, results in the key issue for switch the selectivity towards the achievement of 5-HMF.

## 1. Introduction

The valorization of the biomass is, nowadays, considered an eco-sustainable approach for the production of biofuels and added value platform molecules. Lignocellulosic biomass is principally made up cellulose, hemicellulose and lignin [1]. From the hydrolysis of cellulose and hemicellulose components, it is possible to achieve several platform chemicals that could replace the currently used fossil-based building blocks of well-established chemical processes. In particular, the catalytic conversion of sugars (fructose and glucose) can lead to the formation of furfural alcohol, furfural, 5-hydroxymethylfurfural (5-HMF) or levulinic acid (LA) [2]. Recently, the conversion of fructose to 5-HMF was deeply studied due to its use as starting material of various intermediates as polymer feedstocks, chemicals and alternative fuels (Figure 1) [3]. The significant interest towards the formation and the use of 5-HMF is shown by the increase of literature in the last years [3,4].

During the last few decades, the synthesis of 5-HMF was focused on homogenous acid-catalyzed processes [5,6], or recently, on the use of heterogeneous catalysts under biphasic conditions [3]. In terms of process sustainability, in both cases, different drawbacks occur. The use of an acidic homogeneous catalyst does not allow the recovery of the material and its reuse. Moreover, the presence of free acids induces many side reactions. In fact, 5-HMF easily undergoes rehydration and oligomerization with the formation of formic acid, LA and humins, and as consequence, has a loss in yield of the desired product [7]. An increase of the 5-HMF production is achieved when a biphasic system is used [6]. The organic phase, in fact, allows one to segregate the hydroxymethylfurfural, preventing subsequent reactions (i.e., rehydration to LA). Several organic phases (DMSO, PVP/NMP, MIBK, 2-butanol, ionic liquids) were studied [3], and in particular, when DMSO is used, a higher concentration of furfural is obtained [8,9]. Yilmaz et al. [10] found a 79% fructose conversion and 89% 5-HMF selectivity when the reaction was carried out in DMSO over sulfated silica or Ti-silica catalysts. The substitution of the DMSO with solvents like 2-butanol or methyl isobutyl ketone leads to a 80% selectivity by using HCl in the water phase [11]. Certainly, the use of some organic solvent, such as DMSO, is not affordable for both sustainability and industrial applications. Now, the research is particularly oriented to the use of heterogeneous materials, to the development of efficient catalytic systems and to the optimization of the reaction conditions in order to fulfil the principle of circular chemistry [3]. Several heterogeneous catalystswere reported in recent literature. Moreau et al. studied the fructose dehydration in presence of zeolites in a biphasic system (MIBK:water 5:1) obtaining a 90% of 5-HMF selectivity [12]. The activities of silica, alumina, niobium phosphates, zeolite, MOR-zeolite and amberlyst-15 were tested in water. In their experiments, Ordomsky et al. [13] found a 60% 5-HMF selectivity, when amberlyst-15 and MOR-zeolite were used. The activities of the tested materials was related to both the Lewis and Brønsted acidities; in particular, the Brønsted acidity increased the selectivity towards the formation of 5-HMF; at the same time, Lewis acidity had the opposite effect, inducing side reactions with the consequent formation of carbonaceous species and humins. 

Many other types of catalysts (i.e., tungstated zirconia oxide [14], Cr (III) polyoxometalate [15] and Ag silicotungstic acid [16]) were tested in water. The fructose dehydration was also studied by using solid sulfonic and phosphonic catalysts. In particular, the acid groups were grafted on polyethylene fiber [17], and due to a strong interaction between the acid groups and water, a moderate 5-HMF yield was obtained. Sulfonated mesoporous MCM-41 [18], prepared by H_2_SO_4_ impregnation, induced an almost complete fructose conversion and a 77% 5-HMF selectivity at 190 °C reaction temperature. 

Due to the high value of HMF molecule, (see Figure 1) the investigation on the different catalysts as well as optimum reaction conditions is still in progress. 

On these premises and as a continuation of our interest on the application of solid acid catalysts in the transformation of platform molecules [19,20], the present work deals with the synthesis, characterization and catalytic properties of different solid sulfonic TiO_2_ catalysts on the hydrothermal dehydration of fructose to 5-HMF (Figure 2).

In particular, synthesized ordered TiO_2_ and commercial Evonik Aeroxide P25 were functionalized with a propylsulfonic or sulfonic group directly linked by a grafting procedure, according the procedure described in Figure 3. Once characterized, the materials were evaluated in different reaction conditions by changing the solvents, temperature and catalyst loadings, in order to find the best reaction conditions both in terms of 5-HMF selectivity and sustainability of the process.

The surface and structural properties of the sulfonic titania materials were investigated by means of X-ray diffraction (XRD), N_2_ adsorption isotherms, thermogravimetric analysis (TGA), X-ray photoelectron spectroscopy (XPS) and acid capacity measurements. These characterizations allowed for correlating the physical and chemical catalysts properties with their activity on the conversion of fructose to 5-HMF. 

## 2. Experimental

### 2.1. Materials

Titanium isopropoxide, tetraethyl orthosilicate (TEOS), 3-mercaptopropyltrimethoxy silane (MPTMS), 35% hydrogen peroxide solution, chloro sulfonic acid, fructose and 5-HMF (used as standard) were purchased from Sigma Aldrich (Milan, Italy). Solvents (toluene and dichloromethane) were distilled prior to use.

### 2.2. Sample Preparation

*Synthesis of TiO_2_ support.* Mesostructured titania was prepared according to a previous published procedure [21], by using a non-ionic amphiphilic triblock polymer, pluronic P123, as a template. The polymer was dissolved in 2-propanol containing HCl diluted in water. The mixture was stirred overnight at 35 °C in a 250 mL one neck flask. Ti(i-PrO)_4_ was quickly added to this solution and stirred for 24 h at the same temperature. The molar composition was 1.0 Ti(i-PrO)_4_:34C_3_H_7_O:0.04HCl:3H_2_O:0.02P123. The milky suspension was aged at 40 °C for 24 h in a closed polypropylene chamber. The solid product was filtered; washed with water and then ethanol; and calcined in air at 500 °C for 5 h (ramp of 2 °C/min). 

*Synthesis of sulfonic titania-based materials.* For the synthesis of the catalysts, a grafting procedure was carried out on both commercial Evonik Aeroxide (P25) and synthetized TiO_2_ supports.

*Propyl Sulfonic Acid Titania.* In a typical synthesis by grafting, a mixture of titanium dioxide (2.00 g) in dry toluene (35 mL) and 3-mercaptopropyltrimethoxy silane (MPTMS) (3.33 mmol) was refluxed for 24 h at 110 °C. The material was then recovered by filtration, washed with toluene and ethanol and dried at 120 °C overnight. Thereafter, the mercaptopropyl groups were oxidized to sulfonic groups with 35% hydrogen peroxide solution (2 mL) in methanol (20 mL) at room temperature for 24 h. After filtration, the solid was dried at 80 °C overnight to obtain TiO_2_-PrSO_3_H materials.

*Sulfonic Acid Titania.* In a typical synthetic procedure, chloro sulfonic acid (0.25 mL, 3.6 mmol) was added dropwise to a suspension of TiO_2_ (1.0 g, 1.2 mmol) in dry dichloromethane (10 mL) at 0 °C stirring for 2 h until the HCl gas evolution stopped. The temperature rose to room temperature and the mixture was stirred for 2 h more. Then, the mixture was filtered and washed with ethanol and dried at room temperature to obtain TiO_2_–SO_3_H materials.

### 2.3. Catalyst Characterizations

*X-ray diffraction* patterns were measured with a Bruker vertical goniometer (Billerica, MA, USA) using Ni-filtered Cu Kα radiation. A proportional counter and 0.05° step sizes in 2θ were used. The assignments of the various crystalline phases were based on the JPDS powder diffraction file cards [22].

The *textural properties* were obtained using a Micromeritics ASAP2020 Plus 1.03 (Ottawa, Canada). The fully computerized analysis of the N_2_ adsorption isotherm at 77 K allowed us to obtain, through the BET method in the standard pressure range 0.05–0.3 p/p^0^, the specific surface areas of the samples. The total pore volume, Vp was evaluated on the basis of the amount of nitrogen adsorbed at a relative pressure of 0.998, while mesopore size distribution values and mesopore volumes were calculated by applying BJH model in the range of p/p^0^ of 0.1–0.98.

The *thermogravimetric analyses* of the samples were performed in air using the TGA 1 Star System of Mettler Toledo (Mettler Toledo, Schwerzenbach, Switzerland). About 10 mg of sample was heated from room temperature to 100 °C, left at this temperature for 30 min and then heated to 1000 °C at the rate of 10 °C/min in 40 mL/min of air.

*X-ray photoelectron spectroscopy* analyses (VG Scientific, Sussex, UK) were performed with a VGMicrotech ESCA 3000 Multilab, equipped with a dual Mg/Al anode. The spectra were excited by the unmonochromatized Al K_ source (1486.6 eV) run at 14 kV and 15 mA. The analyzer operated in the constant analyzer energy (CAE) mode. For the individual peak energy regions, a pass energy of 20 eV set across the hemispheres was used. Survey spectra were measured at 50 eV pass energy. The sample powders were analyzed as pellets, mounted on a double-sided adhesive tape. The pressure in the analysis chamber was of the order of 10^-8^ Torr during data collection. The constant charging of the samples was removed by referencing all the energies to the C 1s set at 285.1 eV, arising from the adventitious carbon. The invariance of the peak shapes and widths at the beginning and at the end of the analyses ensured absence of differential charging. Analyses of the peaks were performed with the CasaXPS software. Atomic concentrations were calculated from peak intensity using the sensitivity factors provided with the software. The binding energy values are quoted with a precision of ±0.15 eV and the atomic percentage with a precision of ±10%.

*Acid capacity* of supports was determined by titration with 0.01 M NaOH (aq). In a typical experiment, 0.1 g of solid was added to 10 mL of deionized water. The resulting suspension was allowed to equilibrate, and thereafter was titrated by dropwise addition of 0.01 M NaOH solution using phenolphthalein as the pH indicator.

### 2.4. Hydrothermal Fructose Dehydration 

A stainless steel autoclave hydrothermal reactor (Tefic Biotech Co. Limited, Xi’an, China) containing a 25 mL PTFE chamber was used. In order to achieve the desired constant temperature conditions the reactor was kept suspended in a thermostatic, preheated synthetic oil bath, posed on a heated magnetic stirrer (VWR International s.r.l., Radnor, Pennsylvania, USA). Inside the chamber a volume of 3 mL of an aqueous solution of fructose 1.1 M and 6 mg catalyst load were used along with an equivalent volume of organic phase (7:3 *w*/*w* ratio Methyl-isobuthyl-ketone/2-butanol) [23]. Moreover, a magnetic stir bar turning at 500 rpm ensured a very good dispersion of the catalyst on the two liquid phases. A typical experiment was set for all the different catalysts as a comparable measure of their performances, and it was carried out for a time of 3 h starting once the autoclave had reached the thermal equilibrium; hence, the transient time has not been counted in the reaction time. Once the reaction time was over, the reactor was left to cool down naturally suspended in air, and after that the suspension was filtered through 0.25 µm membranes (HA, Millipore) to separate the catalyst particles. 5-HMF and fructose concentrations before and after the reaction were analyzed by means of a HPLC Dionex UltiMate 3000 equipped with a column Rezex ROA – Organic acid H+ operating at 60 °C and using 0.6 mL/min of a 5 mM H_2_SO_4_ aqueous solution as eluent. A commercial standard of 5-HMF was used in order to identify and confirm its presence in the reacting medium and to calculate its concentration by a calibration curve. Performance of sulfonic titania-based catalysts was evaluated in terms of fructose conversion (%) and 5-HMF selectivity (%), calculated as below.
$$Conversion\;\left(\%\right)=\;\frac{moles\;of\;converted\;fructose}{initial\;moles\;of\;fructose}\;\times 100$$
$$Selectivity\;\left(\%\right)=\;\frac{moles\;of\;5-HMF\;formed}{moles\;of\;converted\;fructose}\;\times 100$$

Once the best performing material was identified, the same reaction was performed by using only water as the solvent; i.e., without the presence of any organic phase. Moreover, the influence on the activity of reaction time, catalyst loading and initial fructose concentration were also studied.

## 3. Result and Discussion

### 3.1. Structural Characterization

The surface and structural properties of the sulfonic titania materials were investigated by means of X-ray diffraction (XRD), N_2_ adsorption isotherms, thermo-gravimetric analysis (TGA), X-ray photoelectron spectroscopy (XPS) and acid capacity measurements in order to correlate the catalytic performance with the structural properties. 

The XRD spectra of the bare titania supports (Figure 4) show that the synthetized TiO_2_ is a pure anatase form, while the Evonik P25 results in a mixture of anatase and rutile structure. 

Both titania supports were then functionalized by sulfonic groups, those or propyl derivatives, and the obtained catalysts were characterized in order to prove the correct functionalization.

In Table 1, the textural properties in terms of specific surface area and pore volumes are listed for the bare Evonik P25 and synthetized TiO_2_ and their acid derivatives along with their acid capacity and the atomic ratio obtained by XPS analysis. 

The textural properties indicate that, although the SS results are very similar among all the catalysts, materials derived from Evonik P25 present a higher volume and dimension pore (10–100 nm) with respect to the catalysts derived from synthesized TiO_2_ (3–10 nm). 

Figure 5 can better explain the textural properties of the materials. The nitrogen adsorption–desorption isotherms of synthesized TiO_2_ and its sulfonic derivatives, show a type-IV shape with an H_2_ type hysteresis loop, which is typical of mesoporous materials with low surface area and presence of ink-bottle pores (see Figure 5a). After the functionalization with sulfonic or propylsulfonic groups, a slight reduction of the amount of nitrogen adsorbed was observed as confirmation of a reduction of both surface area (from 45 m^2^g^−1^ to 37 and 41 m^2^g^−1^, respectively) and pore volume of materials (from 0.1 cm^3^g^−1^ to 0.07 and 0.09 cm^3^g^−1^, respectively) (Table 1). Taking into account the pore size distributions (see Figure 5c), it is possible to observe a bimodal distribution of the pore width of pristine TiO_2_ with a main contribution between 5–9 nm and a less important one between 3–4 nm. The grafting procedure induces a shift between 4–7 nm of the main peak and an increase in the number of pores between 3–4 nm as a consequence of a functionalization that mainly occurs inside the bigger pores.

Concerning the textural properties of Evonik P25 and its derivatives, the nitrogen adsorption-desorption isotherms (Figure 5b) show a type-III shape without hysteresis loop, which is typical of non-porous materials. In this case, after the functionalization with sulfonic or propylsulfonic groups, a marked reduction of the amount of nitrogen adsorbed is observed, as confirmed by the corresponding reduction of BET surface area (from 56 m^2^ g^−1^ to 35 and 39 m^2^g^−1^, respectively). The pristine Evonik P25 presents a wide distribution of the pores, (Figure 5d) that change depending on the type of functionalized group, and in particular, on the different functionalization procedure adopted. In fact, when the sulfonic is grafted a wide distribution is maintained, whereas when the propylsulfonic is considered, a sharper distribution centered at 40 nm is observed. The fact can be explained assuming that the pores present in pristine P25 are due to the empty space formed by aggregation of the spherical particles of the material. When the functionalization with propylsulfonic group occurs, the oxygen evolution, produced during the oxidation step with H_2_O_2_, can cause the partial disaggregation of weakly bound particles, leaving only the strong aggregates that are characterized by smaller pores.

The successful functionalization can be also demonstrated by the thermogravimetric spectra (Figure 6) that indicate notable higher degree of functionalization of the samples prepared over Evonik P25 even though the TiO_2_ samples show a higher thermal stability. 

The XPS analysis (see Table 1) confirms a high presence of the grafted groups in the Evonik P25 based samples, showing atomic ratios S/Ti of 0.58 and 0.43 for the propyl sulfonic and sulfonic derivatives, respectively. When the corresponding samples of the synthesized titania were analyzed, only atomic ratios of S/Ti of 0.19 and 0.17 for the –PrSO_3_H and –SO_3_H derivatives, respectively, were found. 

Finally, according to the results found by both XPS and TGA analysis, the acid capacities of the samples (see Table 1) are higher in those materials coming from the Evonik P25 with respect to those obtained by the synthesized titania. Moreover, as far as it concerns the specificity of the grafted group, the sulfonic derivatives (-SO_3_H) showed a higher acid capacity, determined by titration method explained in the experimental section, (1.60 and 0.30 mmol H^+^g^−1^ for P25 and TiO_2_, respectively) with respect to the propyl sulfonic ones (0.73 and 0.16 mmol H^+^g^−1^ for P25 and TiO_2_, respectively).

### 3.2. Catalytic Activity

The hydrothermal dehydration of fructose was studied in the presence of the different solid acid materials (-PrSO_3_H and -SO_3_H) and the catalytic behavior of the materials was evaluated in terms of both conversion of the starting material and selectivity versus the formation of 5-HMF (Figure 2).

Following recent procedures [20], a biphasic system 1:1 water/organic (3:7 sec-BuOH/MIBK), was used for the screening of the prepared acid catalysts, and the results are summarized in Figure 7. According the used reaction conditions, the effect of the reaction temperature (T 140 °C and 165 °C) on fructose conversion and 5-HMF selectivity was studied. The rise of temperature to 165 °C led to an almost complete conversion (95%–99%) of fructose for all catalysts, although the 5-HMF selectivity was higher for the sulfonic and propyl sulfonic catalysts grafted on synthesized titania. Furthermore, the best result in terms of 5-HMF selectivity (78%) was achieved by using sulfonated titania catalyst (TiO_2_-SO_3_H) at a lower temperature (140 °C). Moreover, the same catalyst gave the best results at 165 °C, showing an almost complete conversion of fructose with a good selectivity to 5-HMF (ca. 66%). For comparison purposes, the reaction was carried out in the absence of the catalitic powder as well, at the reaction temperature of 140 °C by using sulfuric acid solution as homogeneous catalyst (same mmol H^+^ of TiO_2_-SO_3_H catalyst), and despite an almost complete conversion of fructose, the selectivity to 5-HMF fell down to 9%. By comparing the results found with the various catalysts, it is possible to observe that the catalysts with the higher numbers of acid sites (see Table 1) showed the best performances in terms of fructose conversion but the lowest selectivity versus the 5-HMF formation. This behavior is confirmed also by the runs carried out in the presence of sulfuric acid, indicating that exist an optimum in the amount of acid sites in order to convert fructose into 5-HMF with a good selectivity. 

Due to these interesting results, TiO_2_-SO_3_H material, which showed the best compromise between conversion and selectivity, was chosen for a further study on optimization of the reaction conditions. 

In order to make greener the process, fructose dehydration was then studied at the previous two temperatures, 165 °C and 140 °C, by using as the solvent, only water. The results for the runs carried out at 165 °C and at 140 °C are listed in Table 2 and Table 3, respectively. 

By comparing the results obtained in water with those found in the presence of the biphasic solvent, it can be observed that at 165 °C, after three hours of reaction, the conversion was maintained at ca. 100 %, and instead, at 140 °C it was greatly increased (ca. three times). On the contrary, as expected, at both temperature the selectivity versus 5-HMF decreased at ca. 50% when only water was used as solvent. The above results indicate that when the reaction was carried out at 165 °C for 3 h, the total conversion of fructose was always reached, but in water, the maximum selectivity to 5-HMF was 50%. In order to try to increase the selectivity, two runs at 165 °C were also carried out for 1 and 2 h (see Table 2). These runs evidenced that for lower reaction times the conversion decreased, but only a very slight increase of the selectivity was observed. Consequently, it seems that exists a maximum of selectivity equal to 50% at 165 °C. For this reason, a further study was carried out at 140 °C. 

A run carried out in the absence of catalyst, not reported in Table 3, showed a conversion and a selectivity of ca. 20% with a yield to 5-HMF of ca. 4%. This fact suggests that a correct acidity degree of the reaction medium is important in order to obtain not only a good conversion of fructose, but also a good selectivity versus the formation of 5-HMF. For this reason, reactions were carried out with different amounts of catalyst in the suspension. In particular, two tests were performed with 18.0 and 36.0 mg of catalyst, respectively. By increasing the catalyst amount from 6 to 18 mg, all parameters (conversion, selectivity and yield) increased (37%, 59% and 22% respectively), whereas, when a higher amount of catalyst (36 mg) was used, it was found that the selectivity to the desired product decreased to 49%.

The lower 5-HMF selectivity and a slightly higher fructose conversion (42%) can be associated to the presence of undesired products. In fact, the accumulation of fructose on the surface of the material led to a high formation of carbonaceous by-products inducing the darkening of the material and the poisoning of the surface of the catalyst. In order to avoid this problem, a lower fructose concentration (0.1 M) was used, the reaction was carried out during one hour and the results are shown in Table 4. Moreover, because of a lower concentration, the presence of by-products derived from subsequent reactions, was reduced and, at the same catalyst amount (18 mg), a remarkable increase of 5-HMF selectivity (from 59 to 71%) was registered. Once again, the use of a double amount of catalyst (36 mg) does not lead to an improvement of the results; in fact, even though a higher 5-HMF selectivity (88%) was achieved, the conversion (10%) slightly decreased and therefore the yield remains almost equal (ca. 9%). 

Consequently, after the optimization of the reaction conditions, the best compromise, in terms of high 5-HMF selectivity and greener conditions, was found by using 0.1M fructose concentration, in water as solvent (3 mL), at the temperature of 140 °C, in the presence of a catalyst amount of 18 mg for a one hour reaction. 

Through the analysis of the results and the characteristics of the synthesized materials, the catalytic activities of the materials can be related to their physicochemical properties. A higher acidity (0.73 and 1.6 mmol H^+^g^−1^) of Evonik P25 derivatives induces a higher conversion of fructose, but at the same time promotes oligomerization and other reactions for the formation of humins and by-products. Moreover, the higher pore volume (0.30 and 0.38 cm^3^g^−1^) and pore size (10–100 nm) allow a longer build-up of fructose and 5-HMF molecules, and consequently, further side acid-catalyzed processes. On the other hand, the activity of synthesized titania derivatives results in more selection towards the formation of furfural. The weak acidity (0.16 and 0.3 mmol H^+^g^−1^) along with the small pore volume (0.09 and 0.07 cm^3^g^−1^) and its size (3–10 nm), catalyzed the fructose conversion, avoiding the permanence of the desired product inside the pore and further adverse reactions. This insight is also confirmed by the results obtained when a lower concentration of fructose is used. According to these considerations, a plausible reaction pathway for fructose dehydration catalyzed by TiO_2_-SO_3_H is described in Figure 8. The sulfonic groups linked to the OH catalyzed the water removal through three consecutive steps by E1 mechanism: two steps of 1,2-elimination and one step of 1,4-elimination reaction.

## 4. Conclusions

In the present study, two different acid functionalized titanium dioxide powders were used as the catalyst for the conversion of fructose into the platform molecule 5-HMF. During the screening of the materials, the synthesized TiO_2_ directly functionalized with the sulfonic group (TiO_2_-SO_3_H) showed the best activity in terms of selectivity of furfural. Particular attention was paid to the sustainable reaction conditions by using water as solvent and low catalyst/fructose ratio. The TiO_2_-SO_3_H catalyst can be made by one step process, starting from as economic and ecofriendly a material as TiO_2_. In the recent literature, the selected catalyst, TiO_2_-SO_3_H, showed a good activity in terms of temperature, reaction time, conversion and selectivity, with respect sulfonic materials used in similar conditions, such as sulfonic MCM41 [18], Amberlyst-15 [14], HSO_3_-fiber [17]. Moreover, with respect to the use of several metal supported catalyst [3], in terms of sustainability, TiO_2_-SO_3_H material present the advantages of avoiding the use of toxic and barely-available metals in its structure. 

Finally, due to the interesting findings obtained by this study, further investigations will be oriented to the implementation of TiO_2_-based catalysts for the production of 5-HMF using the easily available glucose as starting material. 

## Figures and Tables

**Figure 1 materials-13-01178-f001:**
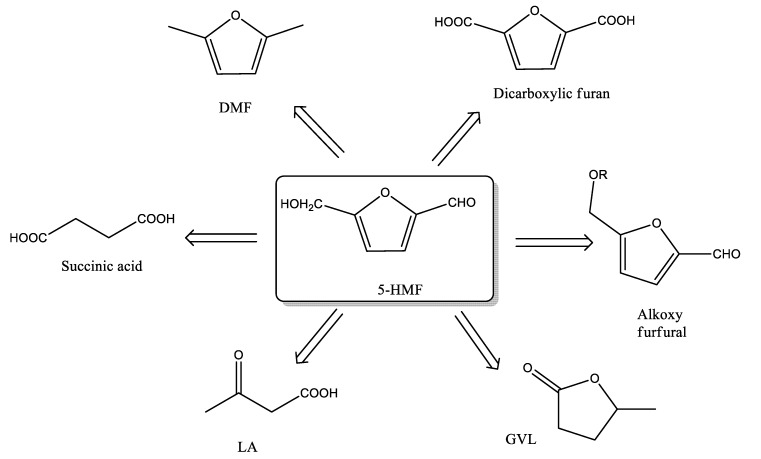
5-HMF derived added value products.

**Figure 2 materials-13-01178-f002:**
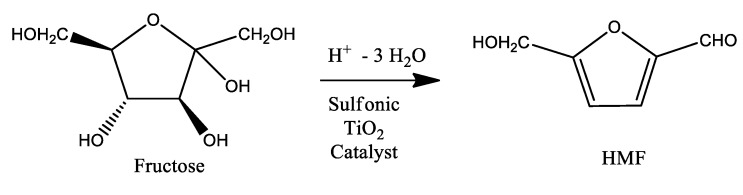
Fructose dehydration to 5-HMF.

**Figure 3 materials-13-01178-f003:**
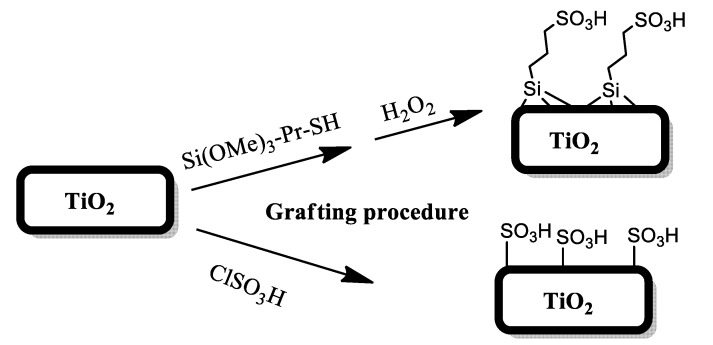
Grafting procedure for the synthesis of TiO_2_ catalysts.

**Figure 4 materials-13-01178-f004:**
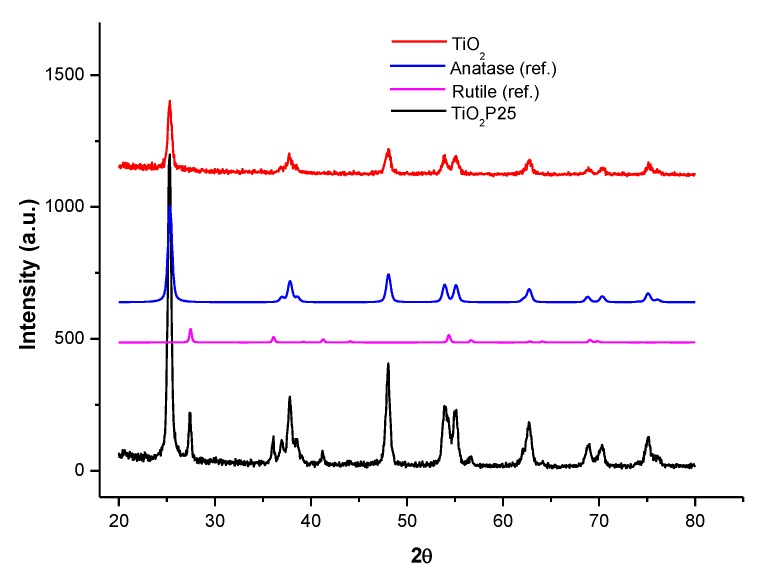
XRD spectra of titania supports.

**Figure 5 materials-13-01178-f005:**
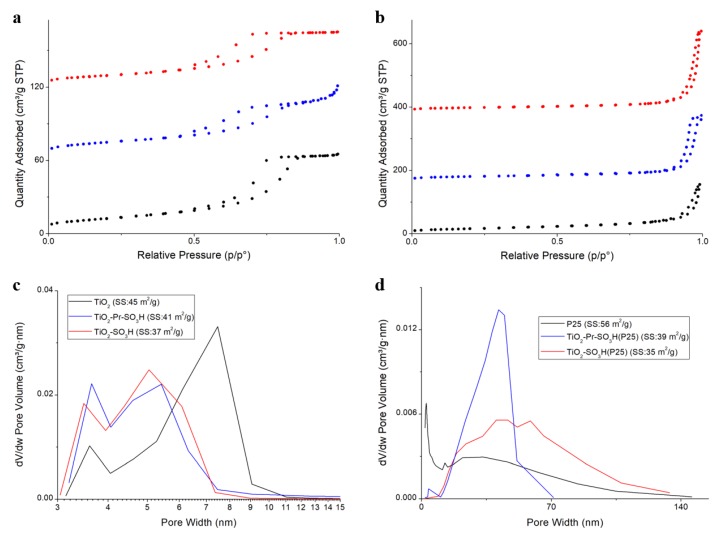
(**a**,**b**) Nitrogen adsorption–desorption isotherms (for sake of clarity the curves are shifted up their y axes), and (**c**,**d**) pore distribution of bare TiO_2_ and its derivatives.

**Figure 6 materials-13-01178-f006:**
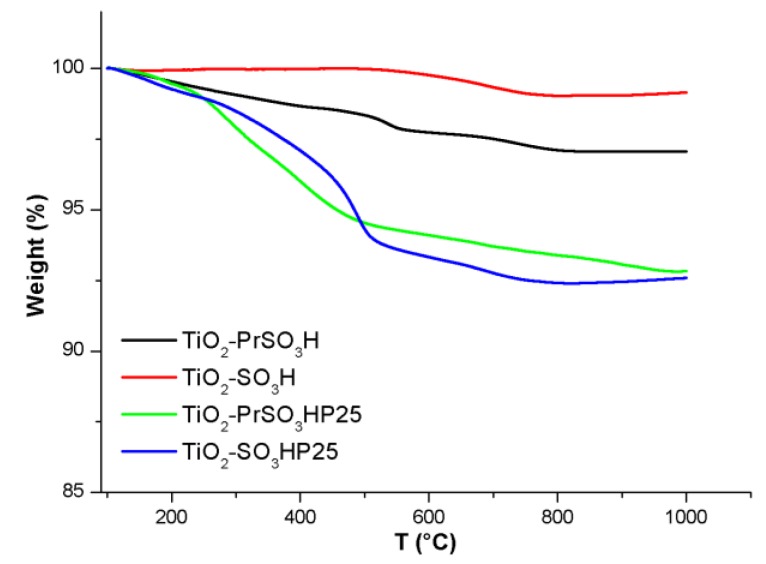
Thermogravimetric analysis of the functionalized TiO_2_ catalysts.

**Figure 7 materials-13-01178-f007:**
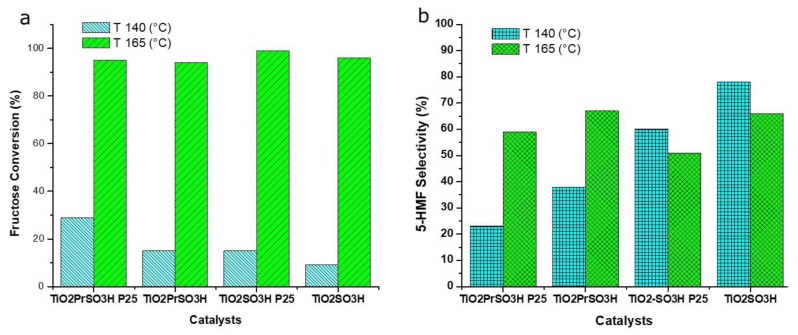
Fructose conversion (**a**) and 5-HMF selectivity (**b**) over acid TiO_2_ catalysts. Reaction conditions: fructose 1.1M, 3 mL biphasic system 1:1 water/organic (3:7 sec-BuOH/MIBK), reaction time 3 h, catalysts 6 mg.

**Figure 8 materials-13-01178-f008:**
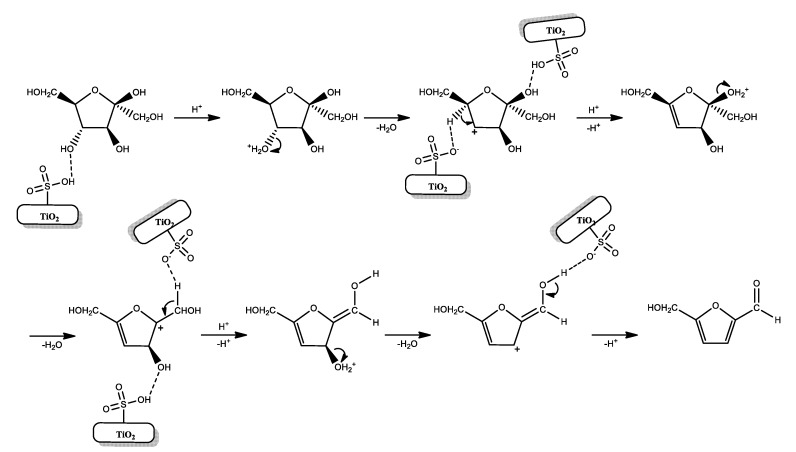
Proposed mechanism of fructose dehydration to HMF over TiO_2_–SO_3_H.

**Table 1 materials-13-01178-t001:** Textural properties, XPS atomic ratio and acid capacity of each prepared material.

Catalyst	BET		XPS			Acidity (mmol H^+^g^−1^)
	SSA (m^2^g^−1^)	Vp (cm^3^g^−1^)	S/Si	S/Ti	Si/Ti	
TiO_2_	45	0.10				0.08
TiO_2_ P25	56	0.24				0.08
TiO_2_-Pr-SO_3_H	41	0.09	0.35	0.19	0.53	0.16
TiO_2_-Pr-SO_3_H P25	39	0.30	0.73	0.58	0.80	0.73
TiO_2_-SO_3_H	37	0.07		0.17		0.30
TiO_2_-SO_3_H P25	35	0.38		0.43		1.60

**Table 2 materials-13-01178-t002:** Time profile of TiO_2_-SO_3_H performance on the fructose dehydration in water.

Time (h)	Conversion (%)	Selectivity (%)	Yield (%)
1	82	53	43
2	89	51	45
3	99	50	50
(*) 3	99	66	65

Reaction conditions: fructose 1.1M, 3 mL water, T = 165 °C. (*) For comparison purposes, results were obtained in a biphasic system.

**Table 3 materials-13-01178-t003:** Effect of the TiO_2_-SO_3_H catalyst amount on fructose dehydration in water.

Catalyst Amount (mg)	Conversion (%)	Selectivity (%)	Yield (%)
(*) 6.0	9	78	7
6.0	25	48	12
18.0	37	59	22
36.0	42	49	20

Reaction conditions: fructose 1.1M, 3 mL water, reaction time 3 h, T = 140 °C. (*) For comparison purposes, results were obtained in a biphasic system.

**Table 4 materials-13-01178-t004:** Effect of the fructose concentration on TiO_2_-SO_3_H catalyst performances.

Catalyst Amount (mg)	Conversion (%)	Selectivity (%)	Yield (%)
6.0	14	31	4
18.0	13	71	9
(*) 18.0	29	55	16
36.0	10	88	9

Reaction conditions: fructose 0.1 M, 3 mL water, reaction time 1h, T = 140 °C. (*) For comparison purposes, results were obtained by using fructose 1.1M.

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
