# Peer review of "Dehydration of Fructose to 5-HMF over Acidic TiO2 Catalysts"

_materials, 2020, doi:10.3390/ma13051178_

Round 1
Reviewer 1 Report
In this work, the authors reported the dehydration of fructose to HMF using the acidic TiO2 catalysts. Different solid sulfonic titania-based catalysts were investigated on the hydrothermal dehydration of fructose to HMF. In comparison with previous works, the data reported here require further evidence and this paper can be published after a major revision. 1. Abstract: The authors should include the main data, especially for the yield of HMF as well as the conversion. This works requires the proper references to support what they said. For example on P1:Line 22-28:“The valorization of the biomass is, nowadays, considered an eco-sustainable approach for the production of biofuels and added value platform molecules. Lignocellulosic biomass is principally made up by monosaccharides such as xylose, fructose and glucose, convertible by catalytic processes to eco-sustainable platform chemicals, that could replace the currently used fossil-based building blocks of well-established chemical processes. By the 26 catalytic conversion of sugars (xylose, fructose and glucose) it is possible to obtain furfural alcohol, 27 furfural, 5-HydroxyMethylFurfural (5-HMF) or levulinic acid (LA). “ The authors did not include any references to support what they said. This is not right. 3. Figure 1 requires the proper references to support the chemicals produced from HMF 4. P2: Line 36-40: “Both 36 cases present several drawbacks in terms of process sustainability. By using a homogeneous catalyst, 37 the material cannot be reused and many by-side reactions occur. In fact, 5-HMF easily undergoes 38 rehydration and oligomerization with the formation of formic acid, LA, humins and, as consequence, 39 a loss in yield of the desired product.” 5. NH3-TPD should provide to measure the exact amount of acidic Ti-support. 6. I suggest the title to be changes into: Fructose Dehydration to 5-HMF over acidic TiO2 catalysts. 7. References should be kept uniform format according to the Guideline.
Author Response
Reviewer #1: (1) Abstract: The authors should include the main data, especially for the yield of HMF as well as the conversion.
The required data were added to the abstract.
(2) This works requires the proper references to support what they said. For example on P1:Line 22-28:“The valorization of the biomass is, nowadays, considered an eco-sustainable approach for the production of biofuels and added value platform molecules. Lignocellulosic biomass is principally made up by monosaccharides such as xylose, fructose and glucose, convertible by catalytic processes to eco-sustainable platform chemicals, that could replace the currently used fossil-based building blocks of well-established chemical processes. By the catalytic conversion of sugars (xylose, fructose and glucose) it is possible to obtain furfural alcohol, furfural, 5-HydroxyMethylFurfural (5-HMF) or levulinic acid (LA). “ The authors did not include any references to support what they said. This is not right.
The introduction was improved with the suggestion of the referee and the corresponding references were properly added.
(3) Figure 1 requires the proper references to support the chemicals produced from HMF
As suggested by the reviewer we added the corresponding reference.
(4) P2: Line 36-40: “Both 36 cases present several drawbacks in terms of process sustainability. By using a homogeneous catalyst, 37 the material cannot be reused and many by-side reactions occur. In fact, 5-HMF easily undergoes 38 rehydration and oligomerization with the formation of formic acid, LA, humins and, as consequence, 39 a loss in yield of the desired product.”
The paragraph was better explained in order to be clearer to readers
(5) NH3-TPD should provide to measure the exact amount of acidic Ti-support.
The NH3-TPD measurement, which, indeed, would be very interesting for the determination of exact amount and strength of acidic groups, is in our case not feasible. As showed in Fig. 6 and found with analogous measure in inert atmosphere, the thermal stability of the propyl sulfonic group does not reach 450 °C. This means that during the desorption step of ammonia we could have a partial or total detachment of the acidic groups. The measure of acidity would then be not indicative of the whole acidity. We think that, in our case, because the surface area of catalysts are comparable, and in view of the fact that the reaction is performed in water, the measure of acidity performed by titration in water, is a good compromise in order to compare our catalysts.
(6) I suggest the title to be changes into: Fructose Dehydration to 5-HMF over acidic TiO2 catalysts.
According the referee‘s suggestion, the title of the manuscript was changed
(7) References should be kept uniform format according to the Guideline.
The references were written according to the Guideline

Reviewer 2 Report
The authors report the use of titanium catalysts for the dehydration of fructose. Initially the surface and structural properties of the catalysts were analysed. Then, the conversion and selectivity of the reaction process are discussed when modifying the reaction conditions.
The manuscript could be accepted after minor revisions as followings.
How authors identify HMF and the by-side products? Did they use commercial standards? No spectroscopic data are included in the manuscript to confirm the HMF structure in the reaction. Some information, NMR data or mass analysis, or alternatively information about the standards should be included, not only HPLC conditions to detect the product. After optimization, authors obtained the bests results using just 0,1M fructose, 3 mL water and 18 mg catalysts, and reaction yield is not very good. Could the reaction be scalable? For instance, in multi-gram scale? On page 2, figure 1,delete ‘amine’ from the molecule drawn in the top right corner due to ‘amine’ it is not the name or acronym of the molecule. On page 10, conclusions, ‘low temperature’ should be removed due to 140ºC implies heating at not so low temperature. The head of tables 2, 3, and 4 are the same. Some additional information could differentiate all of them. The format of references should be checked carefully, in terms of, commas in bold that should not be in bold, missing full stops,…Author Response
Reviewer #2: (1) How authors identify HMF and the by-side products? Did they use commercial standards? No spectroscopic data are included in the manuscript to confirm the HMF structure in the reaction. Some information, NMR data or mass analysis, or alternatively information about the standards should be included, not only HPLC conditions to detect the product.
In order to identify and confirm the presence of HMF commercial standards were used. In the experimental section the information about the standards were included
(2) After optimization, authors obtained the bests results using just 0,1M fructose, 3 mL water and 18 mg catalysts, and reaction yield is not very good. Could the reaction be scalable? For instance, in multi-gram scale?
In this manuscript we focused our study on the optimization and sustainability of the reaction conditions. Certainly, further investigation could be oriented towards the scalability of the reaction, that, in our opinion, is achievable. In fact, in this case we use a simple autoclave that can be easily scaled for the production of HMF in multi-gram amount
(3) On page 2, figure 1,delete ‘amine’ from the molecule drawn in the top right corner due to ‘amine’ it is not the name or acronym of the molecule. On page 10, conclusions, ‘low temperature’ should be removed due to 140ºC implies heating at not so low temperature. The head of tables 2, 3, and 4 are the same. Some additional information could differentiate all of them. The format of references should be checked carefully, in terms of, commas in bold that should not be in bold, missing full stops,…
According all the suggestions of the referee: - Figure 1 was modified in a correct way - In the conclusion the term low temperature was eliminated - The head of Tables 2,3,4 was written related to the data reported. – References were written according to the Guideline

Reviewer 3 Report
More detailed characterisation should be carried out to help understand the catalyst properties and behaviour. The effect of catalyst properties on the catalyst performance is not well understood. I recommend publication after inclusion of relevant characterisation to help elucidate the roles of functional groups.
Author Response
Reviewer #3: More detailed characterisation should be carried out to help understand the catalyst properties and behaviour. The effect of catalyst properties on the catalyst performance is not well understood. I recommend publication after inclusion of relevant characterisation to help elucidate the roles of functional groups.
We understood the suggestion of the reviewer but unfortunately the most relevant characterization that would be useful to correlate catalysts properties with their activity, i.e. the NH3-TPD measurement, cannot be performed on our samples. Indeed, as showed in Fig. 6 and found with analogous measure in inert atmosphere, the thermal stability of the propyl sulfonic group does not reach 450 °C. This means that during the desorption step of ammonia we could have a partial or total detachment of the acidic groups. The measure of acidity would then be not indicative of the whole acidity. We think that, in our case, because the surface area of catalysts are comparable, and in view of the fact that the reaction is performed in water, the measure of acidity performed by titration in water, is a good compromise in order to compare our catalysts. However, further to the activity of the materials, a better explanation of the catalytic behavior of sulfonic groups grafted on titania on the fructose dehydration was written.

Round 2
Reviewer 3 Report
The present work seems acceptable for publication.